# Rotation Invariant Parallel Signal Processing Using a Diffractive Phase Element for Image Compression

Habib Hamam [1,2,3]



1   Faculty of Engineering, Université de Moncton, Moncton, NB E1A3E9, Canada; habib.hamam@umoncton.ca;
    Tel.: +1-506-858-4762
2   Spectrum of Knowledge Production & Skills Development, Sfax 3027, Tunisia
3   Department of Electrical and Electronic Engineering, School of Electrical Engineering, Science,
    University of Johannesburg, Johannesburg 2006, South Africa

**Abstract:** We propose a new rotation invariant correlator using dimensionality reduction. A diffractive phase element is used to focus image data into a line which serves as input for a conventional correlator. The diffractive element sums information over each radius of the scene image and projects the result onto one point of a line located at a certain distance behind the image. The method is flexible, to a large extent, and might include parallel pattern recognition and classification as well as further geometrical invariance. Although the new technique is inspired from circular harmonic decomposition, it does not suffer from energy loss. A theoretical analysis, as well as examples, are given.

**Keywords:** rotation invariance; parallel information processing

## 1. Introduction

Many methods have been used to achieve rotation invariant pattern recognition [1]. Recent methods are based on deep learning techniques [2,3]. The wedge-ring detector method is one approach that also exhibits scale invariance properties [4,5]. An analysis of the different techniques for recognizing and detecting objects under extreme scale variations were presented by [6]. One of the frequently used filters is the circular harmonic filter (CHF). The correlation using such a filter is invariant under the rotation of the scene image but suffers from the defects that result from only one circular harmonic component (CHC) being used. As a result, the correlation peak is not sufficiently sharp.

To relax such a limitation, various methods have been proposed. These attempts are still the object of active research. However, one should emphasize that the critical parameters of optical efficiency, the sharpness of correlation peaks, the peak-to-sidelobe ratios, resistance to noise, and circular harmonic techniques yield as good or better results than competitive methods. This paper presents a rotation invariant approach based on the optical lossless implementation of one circular harmonic component by means of a diffractive optical element. Both the scene image and reference are subjected to a projection onto one CHC. For clarity, we will use the zero-order CHC, but other CHCs can be used as well.

The diffractive element sums information over each radius of the scene image and projects the result onto one point of a line located at a certain distance behind the image. The method is, to a large extent, flexible and might include parallel pattern recognition and classification as well as further geometrical invariance. The database of references (of fingerprints of subscribers in a bank, for example) is compressed.

The contributions of this paper may be summarized as follows.

*   Compression is performed in a way that the rotated images give the same compressed data.

- In contrast to CHC filters, rotation invariance is ensured without any significant energy loss.
- By maintaining rotation invariance, the image compression technique allows parallel data processing.
- The proposed method might be used to further add geometrical invariance, such as scale invariance, given that during the compressing task a scale factor can be easily included by means of the diffractive compressing element.

The remainder of the paper is organized as follows. Section 2 provides an overview on related works. Section 3 presents a mathematical analysis of the issue of rotation invariance. Section 4 proposes a possible optical implementation. An extension of the proposed architecture is given in Section 5. Results are presented in Section 6. Finally, Section 7 presents concluding remarks.

## 2. Related Works

Full rotation invariance can be ensured by using a matched filter built from a CHC of the reference [7]. In other words, the reference is replaced by one of its CHCs. To improve the peak sharpness and/or the discrimination ability of the classical CHF, several designs based on CHCs, such as the CH covariance filter [8], the phase-only CHF [9], and the phase-derived CHF [9] were proposed. These attempts of improvement, however, were possible at the cost of a decrease in the signal-to-noise ratio (SNR). It is fair to say that the oldest design, the classical CHF, yields the highest SNR among the CHF family. In addition, the classical CHF maximizes the SNR while maintaining the in-plane rotation invariance [10].

The SNR decreases as the order of the CHC is increased. The zero-order CHF is the best choice for pattern recognition under noisy conditions [10]. Unfortunately, low-order CHFs have a tendency to produce broad correlation peaks. This limitation can be overcome, and we can achieve, on the one hand, an important noise resistance, and on the other hand, sharp correlation peaks in addition to high optical efficiency. The technique consists of projecting both the scene image and the reference onto the zero-order CHC. The discrimination ability remains unchanged. The low discrimination ability of the CHFs is a direct consequence of image compression. The proposed approach can be combined with various design techniques to improve this criterion. However, this is not the objective of this work. The resulting image compression into one dimension can be extended to include parallel image processing and further geometrical invariance, as well as by using the second spatial dimension. An optical setup, including a bank of one-dimensional patterns, is proposed for parallel classification, where a dataset of scene images is entered simultaneously.

## 3. Analysis

A circular harmonic filter is one component from the circular harmonic expansion [7]:

$$f(r, \theta) = \sum_{m=-\infty}^{+\infty} f_m(r) exp(im\theta) \tag{1}$$

with

$$f_m(r) = \int_0^{2\pi} f(r, \theta) exp(-im\theta) d\theta \tag{2}$$

The zero-order CHC is merely the sum of information over each ring with radius $r$:

$$\widetilde{f}(r) = f_0(r) = \int_0^{2\pi} f(r, \theta) d\theta \tag{3}$$

Rotated images have the same zero-order CHC. When we apply the correlation operator between the zero-order CHCs of the image $g$ and the reference $f$, a correlation peak is obtained if the input scene image is a rotated version of the reference, including rotation with zero degrees. We then produce a new input scene image $g$ and a new reference $\widetilde{f}$, which are the zero-order CHCs of $g$ and $f$. The expected correlation peak must be sharp because, in contrast to the conventional method, we do not select one component among an infinite expansion. Apart from the fact that both the input scene image and reference are substituted by two derived images, conventional correlation is applied which should give a sharp correlation peak. Moreover, various correlation methods can be applied in conjunction with this image compression.

What is really performed is the following operation, $\widetilde{g} \otimes \widetilde{f}$, where $g$ and $f$ is the original input scene image and reference, and $\otimes$ denotes the correlation operator. The new reference $\widetilde{f}$ is calculated numerically, whereas $\widetilde{g}$ is provided optically. The expected correlation peak arises on the optical axis because both $g$ and $f$ are projected onto the zero-order CHC. In practice, this peak is likely to be hidden by the zero order of diffraction, a useless bright spot on the optical axis. This diffraction order generally results from uncertainty in the fabrication process. To overcome this serious problem, we can laterally shift $\widetilde{f}$ by a certain amount $(x_0, y_0)$. The correlation peak is therefore laterally shifted by the same distance $(x_0, y_0)$, and the detector can be fixed in the same place for all, because its position is known a priori.

In the most general case, the zero-order CHC of the reference is numerically calculated. The lateral shift is also integrated in the numerical procedure. The main difficulty lies in the optical implementation of the projection of the scene image onto its zero-order CHC.

It is worth noting that image data looks squeezed or collapsed into a line that contains all the information of the image scene. This is different from the conventional concept of image compression. This is a result of the fact that rotation invariance is combined with image compression.

## 4. Optical Implementation

Inspired from Equations (1)–(3), we intend to produce the ring-to-point transformation illustrated in Figure 1 (do not consider dashed lines). The objective is to optically implement the integral of Equation (3). All information over each ring of the radius $r$ is summed and the result is projected onto a point shifted by $r$ from the optical axis in a response plane located at a distance $z$ behind the object. The result of the transformation is, thus, a line segment which can be freely oriented in the response plane. For a given radius $r$, the rays covering the optical paths $s(r,\theta)$ must arrive to the collecting point with the same phase, that is, the phase corresponding to $s(r,0)$. To satisfy this constraint, we have to introduce an additional phase distribution $p(r,\theta)$ in the plane of the object fulfilling the following condition:

$$p(r,\theta) + \frac{2\pi}{\lambda} s(r,\theta) = p(r,0) + \frac{2\pi}{\lambda} s(r,0) + 2k\pi \tag{4}$$

where $k$ is an integer and $s(r,0) = z$.

By choosing, for instance $p(r,0) = 0$, the required phase distribution is expressed as follows (see Figure 1):

$$p(r,\theta) = \frac{2\pi}{\lambda} z - \frac{2\pi}{\lambda} \sqrt{z^2 + 4r^2 sin^2\left(\frac{\theta}{2}\right)} + 2k\pi \tag{5}$$

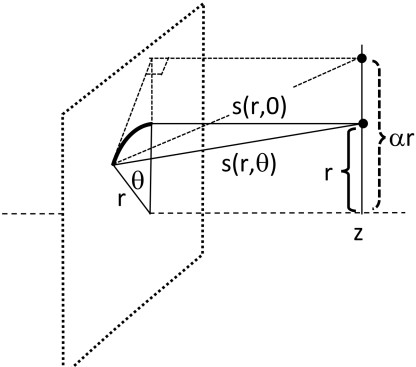

**Figure 1.** Rotation invariant image compression: Ring-to-point transformation.

In the framework of the paraxial assumption, we can use the following approximation ($\sqrt{1 + x^2} \cong 1 + \frac{x}{2}$ if $x \ll 1$. This approximation is used in the Fresnel transform [11,12]):

$$\sqrt{1 + \frac{4r^2}{z^2} sin^2\left(\frac{\theta}{2}\right)} \cong 1 + \frac{2r^2}{z^2} sin^2\left(\frac{\theta}{2}\right)$$

and the required transmittance $t(r,\theta)$ is obtained (for $k = 0$):

$$t(r, \theta) = exp(ip(r, \theta)) = exp\left(-4i\pi \frac{r^2}{\lambda z} sin^2\left(\frac{\theta}{2}\right)\right) \tag{6}$$

We notice that if the implementation of the m-order CHC is required, then we need only to add the term exp(-i $m\theta$) in the expression (6). For instance, one can use the second-order CHF ($m = 2$) which is often a good compromise for both an acceptable SNR and the peak sharpness of the correlation peak [10]. It is also possible to combine several CHCs. This goes, however, beyond the scope of this work.

We need a diffractive phase element with the transmittance expressed by relation (6), which we refer to as the "diffractive compressing element (DCE)". This Fresnel diffractive element has a continuous phase profile (kinoform) [13]. Figure 2 shows a quantized version of such an element, where only four phase levels are used. We projected the phase profile onto the closest phase level in the set {0,$\pi$/2, $\pi$, 3$\pi$/2}. We see in the neighborhood of the bottom half of the vertical axis that the profile is similar to that of a one-dimensional Fresnel lens. Over each ring $r$ of the diffractive element, the phase corresponds to a two-dimensional off-axis Fresnel lens, where the off-axis translation corresponds to $r$. In order to decrease the resolution needed by the Fresnel diffractive element, we can put a spherical lens beside the diffractive element [14,15]. In fact, the distribution of DCE becomes the difference between Figure 2 and the distribution of the inserted lens. It is worth noting that the phase pattern becomes more and more dense in the boundary areas. The smallest feature should be large compared to the wavelength (let us say, more than five times) so that we could remain in the scope of scalar diffraction.

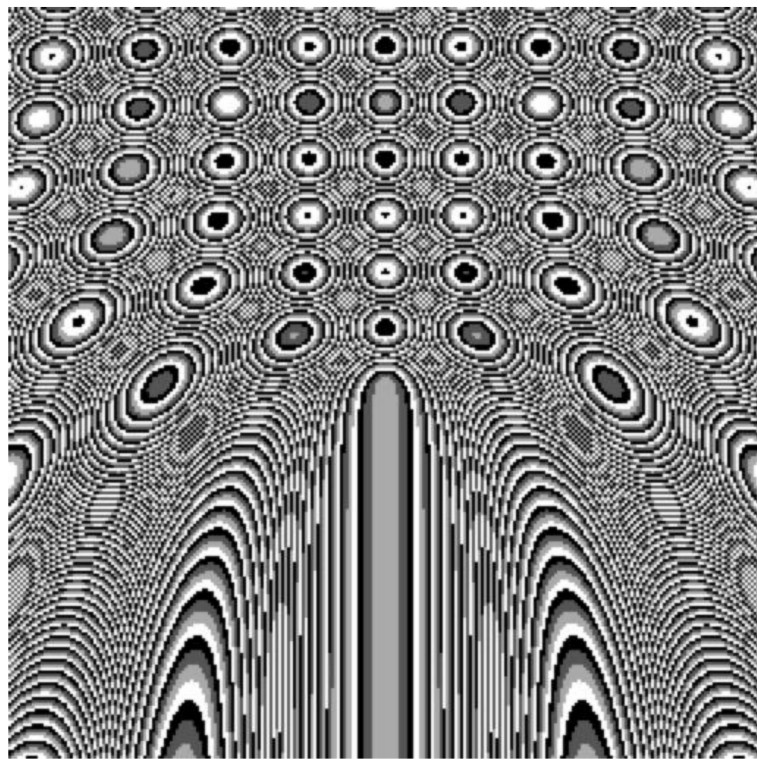

**Figure 2.** Diffractive compressing element (DCE) with four phase levels.

Figure 3 shows the setup of the correlator providing $\tilde{g} \otimes \tilde{f}$. The implementation of the projection of the input scene image is provided by a diffractive phase element placed just behind the input image. The result of the ring-to-point projection is observed at a distance $z$ in the plane $P_C$. The Fourier plane $P_F$ presents, as usual, the filter plane. Using the same diffractive phase element, the filter can be optically implemented.

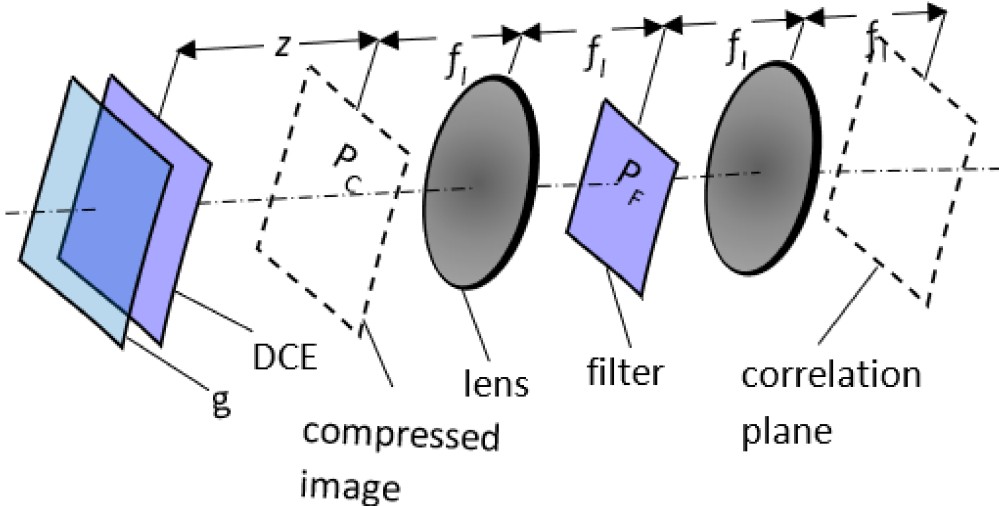

**Figure 3.** Setup of the correlator using a diffractive phase element for image compression. DCE: diffractive compressing element, g: input image.

The DCE is a Fresnel diffractive phase element. It must have a resolution at least as high as the resolution of the input image. The smallest feature of the input image must correspond to at least one phase value of the DCE so that the projection into the line

segment is correctly performed (Figure 1). To obtain better results, the resolution of the DCE must be higher than that of the input image.

## 5. Extensions of the Architecture

### 5.1. Parallel Pattern Recognition

Scale considerations can be taken into account. For instance, we can sum information over each ring of radius $r$ and project the result in a point shifted by the distance $\alpha\,r$ (instead of $r$) from the optical axis, yielding:

$$t(r,\theta) = exp\left(-i\pi\frac{r^2}{\lambda z}\left(1+\alpha^2-2\alpha\cos(\theta)\right)\right) \tag{7}$$

Here we used the Law of Cosines, also known as Al-Kashi's theorem (see dashed lines in Figure 1). This technique can be used, for instance, to adapt the compressed data to the features of the spatial light modulator placed in the Fourier plane.

Owing to the image information compression, it is also possible to provide parallel pattern recognition. For each input scene image $g_m$ we add in the response plane of the diffractive phase element and in the plane of the compressed data $P_C$, a linear phase distribution with a certain slope $exp\left(-i2\pi\frac{x_m r}{\lambda f_l}\right)$; $f_l$ is the focal length of the lens used for the correlation. The Fourier transform of the resulting line segment, containing compressed image data, is laterally shifted by the amount $x_m$ in the filter (Fourier) plane $P_F$. The different slopes can be generated by an array of mini prisms placed in the plane $P_C$. An alternative consists of integrating the slopes in the diffractive element. Therefore, for each input image $g_m$, we need a diffractive phase element with the transmittance:

$$t_m(r,\theta) = exp\left(-i\pi\frac{r}{\lambda}\left(\frac{r}{z}\left(1+\alpha^2-2\alpha\cos(\theta)\right)+\frac{2x_m}{f_l}\right)\right) \tag{8}$$

We need a bank of references where each elementary reference $f_m$ is compared to one input image $g_m$ (Figure 4a). The two-dimensional input images are arranged in a matrix form. After compression by means of the diffractive compressing element DCE, we obtain, in the plane $P_C$, an array of line segments $\widetilde{g}_m$ ($m=1, \ldots , M$). In the Fourier plane, each one-dimensional structure $\widetilde{G}_m$, i.e., the Fourier transform of the compressed image $\widetilde{g}_m$, is multiplied by the conjugate of the Fourier transform $\widetilde{F}_m^*$ and of the compressed pattern $\widetilde{f}_m$ of a two-dimensional reference $f_m$. The Fourier transform, as well as the compression of each reference $f_m$, are performed numerically.

Instead of the output lens $L_2$, we can use the subsystem of Figure 4b, which provides a one-dimensional Fourier transform, noted by $FT_y$. The focal length of the lens placed in the middle of the subsystem of Figure 4b is twice as big as the focal length $f_l$ of the two other identical lenses. The incident wavefront is imaged with respect to one dimension and is Fourier-transformed with respect to the other. We note that the various input scene images can be illuminated by spatially incoherent monochromatic sources, such as a matrix of vertical-cavity surface-emitting lasers (VCSELs). The mutual spatial coherence of the sources is not necessary because the correlation products are performed independently.

Using the same technique of Figure 4, the scale and rotation invariance can be combined. In this case, several replicas of the input image $g$ are entered simultaneously, and for each replica, we attribute a scale factor $\alpha_m$ and a translation factor $x_m$. Therefore, for each replica, we need a diffractive phase element:

$$t_m(r,\theta) = exp\left(-i\pi\frac{r}{\lambda}\left(\frac{r}{z}\left(1+\alpha_m^2-2\alpha_m\cos(\theta)\right)+\frac{2x_m}{f_l}\right)\right) \tag{9}$$

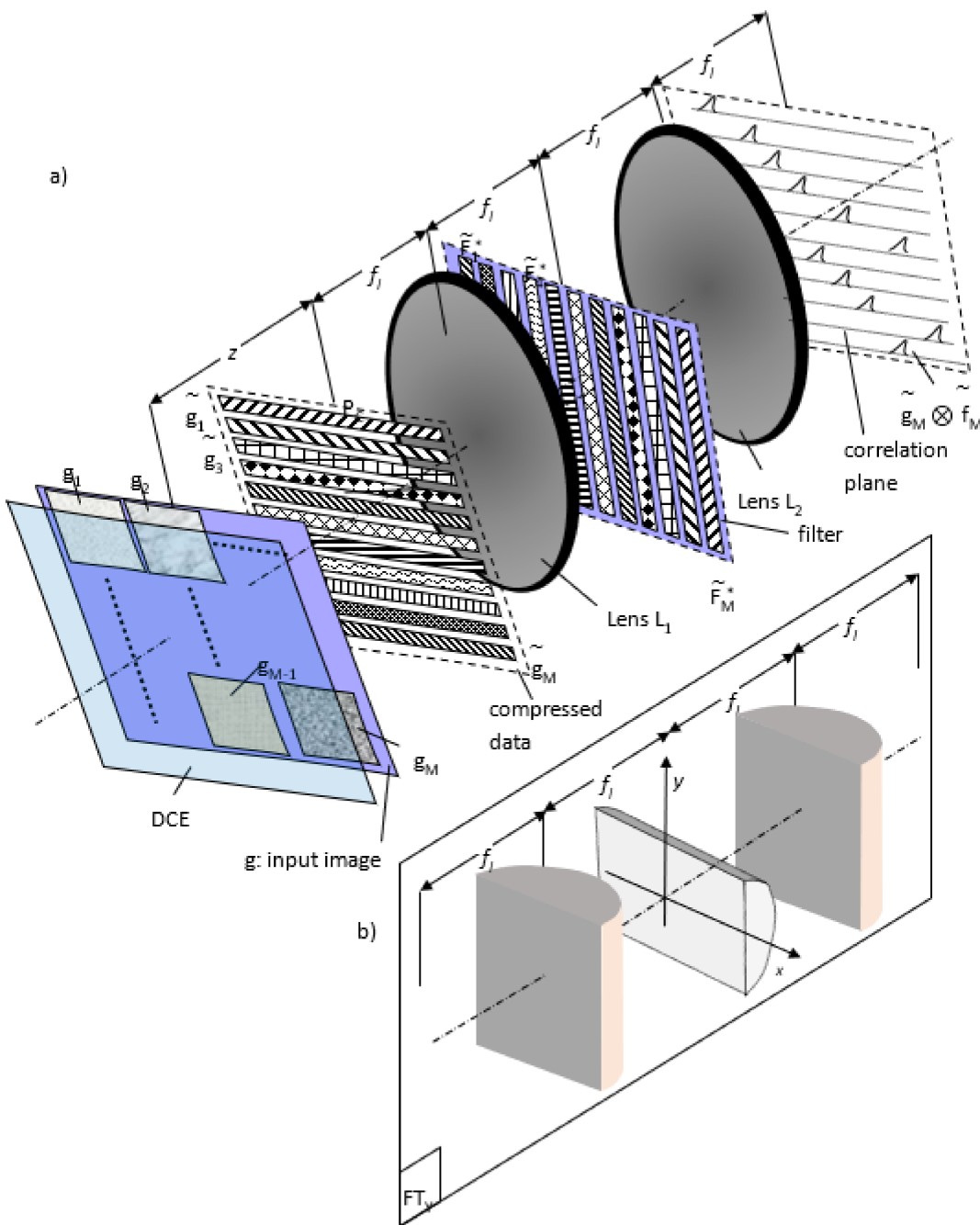

**Figure 4.** Parallel pattern recognition using a bank of one-dimensional patterns (**a**) setup, (**b**) subsystem which provides a one-dimensional Fourier transform according to the y-axis. The focal length of the lens placed in the middle of the subsystem is twice as big as the focal length of the two other identical lenses.

The filter bank, which is the rotation invariant, possesses a two-dimensional structure (Figure 4) where each line segment $\widetilde{F}_m^*$, a conjugate of the Fourier transform of $\widetilde{f}_m$, corresponds to a certain scale $\alpha_m$. The position of the correlation peak is determined by the position of $\widetilde{g}_m$ and that of $\widetilde{f}_m$.

The shift invariance is added to the system by sampling the intensity of the Fourier transforms of the input objects $g_m$, for instance, in combination with a liquid crystal light valve [16]. In other words, what will be calculated is the correlation product of the CHCs of $|G_m(u,v)|^2$ and $|F_m(u,v)|^2$ instead of the CHCs of $g_m(x,y)$ and $f_m(x,y)$.

An alternative for providing scale and rotation invariance consists of entering the input scene image (which is not replicated) and changing the transmittance of the diffractive compressing element. This element must possess the sum of all transmittances (9):

$$t(r,\theta) = \frac{1}{\sqrt{M}} \sum_{m=1}^{M} \left( -i\pi \frac{r}{\lambda} \left( \frac{r}{z} \left( 1 + \alpha_m^2 - 2\alpha_m \cos(\theta) \right) + \frac{2x_m}{f_l} \right) \right) \tag{10}$$

where $M$ is the number of the required scale factors.

In general, the distribution of Equation (10) is not a phase distribution. A projection onto the phase distribution set must be undertaken and the lateral shifts $x_m$ must be correspondingly optimized.

### 5.2. Parallel Pattern Classification

According to Figure 4, each input image is correlated with one filter of the bank. The setup can be modified so as to perform parallel classifications of the input images. Each input image $g_m$ must be compared to all filters $f_m$ of the reference bank. For this purpose, the Fourier transform of each one-dimensional structure $\widetilde{g}_m$ must be replicated in the filter plane. In Figure 5, this operation is performed by a one-dimensional Fourier transform noted by FT$_x$. Each one-dimensional distribution $\widetilde{G}_m$, extended over the y-axis, is Fourier-transformed with respect to the x-axis. The distribution is then replicated to form a two-dimensional structure.

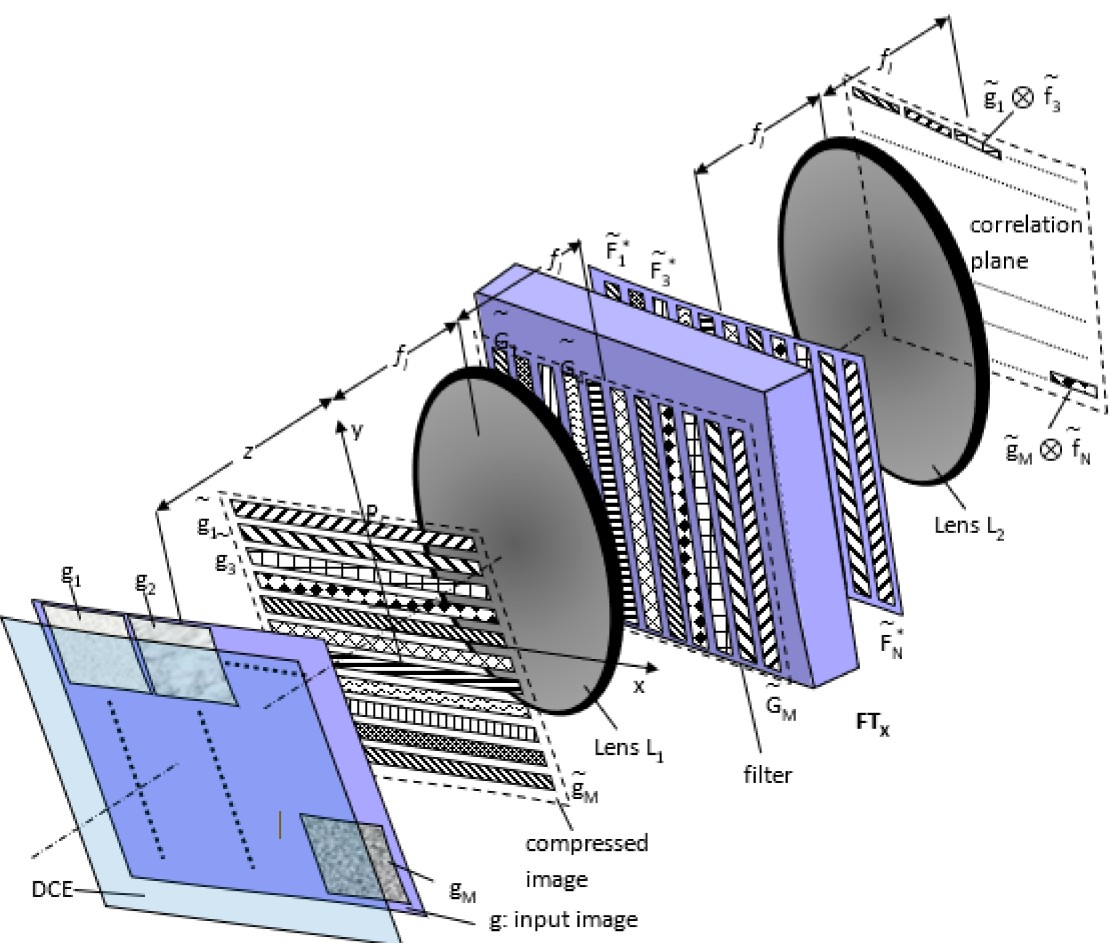

**Figure 5.** Parallel image classification using a bank of one-dimensional patterns, TFx: one-dimensional Fourier transform with respect to the x-axis.

After compressing the input images by means of a multi-faceted diffractive compressing element (DCE), a two-dimensional Fourier transform is performed by means of a spherical lens $L_1$ (Figure 5). Then, the resulting spectra are replicated by a subsystem similar to that of Figure 4. Each replica of one spectrum is multiplied by one one-dimensional pattern from the filter bank $\tilde{F}_1^*$ to $\tilde{F}_N^*$. The correlation product, observed in the focal plane of the spherical lens $L_2$, contains the Fourier transforms of all the products: $\tilde{G}_1\,\tilde{F}_1^*$, $\tilde{G}_1\,\tilde{F}_2^*$, $\dots$, $\tilde{G}_M\,\tilde{F}_N^*$.

The setup of Figure 5 allows for the parallel classification of $M$ input images $g_m$, where a bank of $N$ references, $f_m$, is used.

Rotation invariance is ensured, and scale invariance can be added by enlarging the reference bank. To add shift invariance, the intensity distribution of the Fourier transforms of the objects is taken as the input of the classification system.

## 6. Results

We focus our attention on rotation invariance. The input scene images are presented in Figure 6A. The reference image is shown in Figure 6A-a. Figure 6A-b,A-c are slightly laterally shifted and rotated versions of the reference image, whereas Figure 6A-d is a false image. The energy is normalized for the four images. To test the approach under noisy conditions, significant noise is voluntarily added to the input image. The noise energy is four times bigger than the energy of the useful signal. We designed a binary phase-only filter by projecting the phase profile onto the closest phase level among $\{0, \pi\}$. These kinds of filters can be implemented by a spatial light modulator if a programmable system for pattern recognition is required.

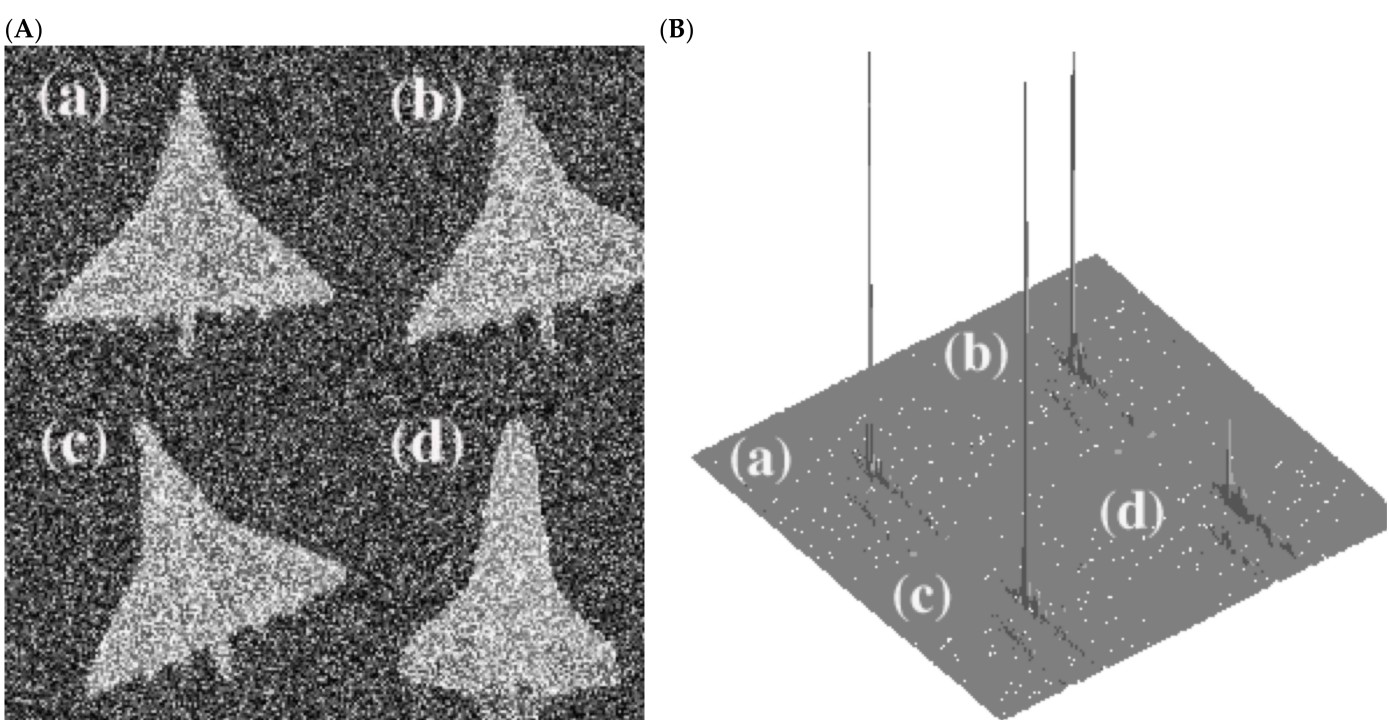

**Figure 6.** (**A**) Input images (a) reference (b) and (c) rotated version of the correct image (d) false image (**B**) Response of the correlator for the four images in (**A**).

The simulation results on Matlab show that rotation invariance is ensured. Figure 6B also shows a high optical efficiency and a good correlation peak sharpness. The optical efficiency is defined as the amount of input light that will be detected for the determination of the correlation function. It is quantitatively measured by the Horner efficiency [17,18]. The energies of the peaks associated to the rotated images are less than that corresponding to

the reference. The approach is sensitive to lateral shift and gives drastic results if the lateral shift is in the range of the image size. Thus, it is necessary, for instance, to use the Fourier transform of the input image to ensure shift invariance.

The diffractive compressing element was quantized onto four phase levels: $\{0, \pi/2, \pi, 3\pi/2\}$. Quantization has been performed by a simple projection onto the closest phase level. Because of quantization we noticed a non-negligible energy loss and it is worth using optimization methods [15].

## 7. Conclusions

The method described here is based on dimensionality reduction by means of image compression. This compression is performed in a way that the rotated images give the same compressed data. In practice, this operation is implemented by a diffractive phase element referred to as a diffractive compressing element. Diffractive phase elements become more and more attractive, mainly because of the technological progress which has especially influenced the fabrication of high resolution diffractive optical elements [19,20]. In our case, we can use a high resolution DCE to improve the discrimination ability. Indeed, the input image is divided in rings with smaller widths, and therefore less compression is performed.

In contrast to CHC filters, rotation invariance is ensured without any significant energy loss. The method makes the optical implementation of the filter possible as well. By maintaining rotation invariance, the image compression technique allows parallel data processing. This parallelism might be used for simultaneous pattern classification. Moreover, it might be used to further add geometrical invariance, such as scale invariance, given that during the compressing task a scale factor can be easily included by means of the diffractive compressing element. For technological reasons, this scale factor can be also used to fit the practical features of the filter, especially when spatial light modulators are used.

However, because of image compression, we obtain a relatively low discrimination ability in practice. Different images might have a similar zero-order circular harmonic component. An alternative might be the extension of the approach into the use of the Fresnel transform-based correlator [21].

**Funding:** The author thanks Natural Sciences and Engineering Research Council of Canada (NSERC) and New Brunswick Innovation Foundation (NBIF) for the financial support of the global project. These granting agencies did not contribute in the design of the study and collection, analysis, and interpretation of data.

**Institutional Review Board Statement:** Not applicable.

**Informed Consent Statement:** Not applicable.

**Data Availability Statement:** Not applicable.

**Conflicts of Interest:** The author declares no conflict of interest.

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
