# Peer review of "Rotation Invariant Parallel Signal Processing Using a Diffractive Phase Element for Image Compression"

_applsci, doi:10.3390/app12010439_

Round 1

Reviewer 1 Report

This manuscript presents a technique that optically “compresses” image data into a line, and feeds the compressed signal to a conventional correlator for classification. The proposed diffractive element can ensure rotation invariance, which means a rotated version of the same image results in the same compressed data. By careful design, parallel processing is feasible. The results demonstrate the effectiveness of the method. The proposed principle is novel, and could find applications in optical pattern recognition etc.

I have the following technical questions and comments for the author to consider.

Image compression is usually used in image processing for the purpose of reducing the cost of storage or transmission. See https://en.wikipedia.org/wiki/Image_compression. However, in this manuscript, the image data is “squeezed” or “collapsed” into a line that contains all the information of the image scene. This is different from the conventional concept of image compression. I suggest finding an alternative expression for this idea in the title and throughout the paper to avoid confusion.

In the introduction, it is not clear how the proposed method is related to image classification and recognition. I think there should be a brief description to link the method to its applications.

Although the technical concept and presentation are easy to understand, the figures in the manuscript have to be significantly improved before publication. In Fig. 1, please consider adjusting the fonts for better display. In Fig. 2, it is better to color code the phase with a decent color map, and display the color bar with labels showing the value range. I assume this is for [0, 2pi] phase, but not clear in its current shape. The optical system in Fig. 3 has a skewed layout, which is not necessary. It is easier to understand if it is aligned horizontally. It is possible to combine Fig. 6 and Fig. 7 into a single figure, as they are both for experimental results.

References: although I appreciate the author listing related papers to this submission, there are some unrelated papers that are from the same author(s). For example, Refs 3 and 4 can be left out, since the proposed method has little relation to COVID-19 diagnosis. On the other hand, there should be some references to the Equations used. For example, Eqs. (1-3) lack proper citations.

There are also some technical issues listed below.

  1. In the second paragraph on pp.3, the correlation operator looks exactly as a multiplication. Conventionally correlation can be expressed with \bigotimes, or corr(f, g).
  2. It is not clear what kind of approximation is used for Eqs. (5-6). Is it Fresnel approximation, or something else? It is important to know the validity of these equations.

  1. In Eq. (6), it should be -2 instead of -4, after applying the approximation and substituting the expressions.

(4) It is obvious in Fig. 2 that there is severe aliasing effect. This raises a question about sampling – is here a limit for the maximum size of the diffractive element in this application. Theoretically the sampling of the phase should comply with the Nyquist sampling rate. This could impose a limit on the overall size of the design. This should be discussed around Fig. 2.

(5) In the paragraph after Fig. 2, it is mentioned that one can put a lens beside the element. So if the approximation in Eq. (6) is Fresnel diffraction, putting a lens will make it a Fraunhofer diffraction. This should be pointed out here. Are they the same to produce the results?

(6) What is the resolution of the proposed DCE?

(7) It is not clear how Eq. (7) is derived. Please describe the process. Is there any approximation here?

(8) In Fig. 4, Lens 2 is not denoted.

(9) In the first paragraph in Sec. 5.2, “each” should be “Each”.

(10) The figure caption after this paragraph should be deleted.

(11) Are the results shown in Sec. 6 in simulation or experimental? It is not clear. What is the designed phase used here? There should be a figure for the phase at least. If it is experimental, a prototype layout is helpful to understand it better. If this is simulation, why there is no experimental results? I think the author should elaborate on this to explain the results in a better and understandable way.

(12) In the paragraph below Fig. 7, the figure number is missing.

Overall, I think the manuscript needs to be revised to address the above concerns before being accepted.

Reviewer 2 Report

Review report of Rotation invariant parallel signal processing using a diffractive

phase element for image compression

In this manuscript, the author presents a new rotation invariant correlator using dimensionality reduction by means of a diffractive phase element. The information over each radius of the scene image is summed by the diffractive element, and the result is projected onto one point of a line located at a certain distance behind the image. Compared to existing rotation invariant pattern recognition methods, the proposed technique in this paper is largely flexible, which achieves an important noise resistance, sharp correlation peaks, and high optical efficiency. The author clearly lists the contributions of the article and expatiates the work from several sections, such as related work, mathematical analysis, and optical implementation, etc. Furthermore, the result demonstrates that the proposed method ensures the rotation invariance and has strong noise resistance and good correlation peak sharpness.

Overall, this paper is well-organized and full of content, which allows the reader to easily get the main idea of the paper. But there are some minor issues needed to be solved for publishing on Applied Sciences.

  1. An equation in Figure 1 is faulty in typography and revision is suggested.
  2. The figure legend “Figure 5. Parallel image classification using a bank of one-dimensional patterns, TFx: one-dimensional Fourier transform with respect to the x-axis.” in section 5.2 is redundant and the author is advised to double-check the whole text to avoid similar problems.
  3. The caption of Fig. 5 is shown after the first paragraph of subsection 5.2, which is redundant and it should be deleted.
  4. In order to demonstrate the accuracy of the extended architecture better, it is suggested that the experimental results about parallel pattern recognition and parallel image classification should be added in this paper.

Round 2

Reviewer 1 Report

The authors addressed most of my concerns in the previous review. I think Fig. 2 is still not clear enough for the readers to understand. It is highly recommended to use a better color map and color bar to denote what is going on here. What's more, the author did not answer that the limitation is. Since the phase pattern becomes more and more dense in the boundary areas, there should be a limit for the largest achievable size for such designs. What is this limit? This is an important question to be answered to understand and pros and cons of the proposed method. 

Author Response

see attached document
